# Extracellular Glycolytic Activities in Root Endophytic *Serendipitaceae* and Their Regulation by Plant Sugars

**DOI:** 10.3390/microorganisms10020320

**Published:** 2022-01-29

**Authors:** Vincenzo De Rocchis, Thomas Roitsch, Philipp Franken

**Affiliations:** 1Leibniz Institute of Vegetable and Ornamental Crops, Theodor-Echtermeyer-Weg 1, 14979 Großbeeren, Germany; 2Institute of Biology, Humboldt-Universität zu Berlin, Philippstrasse 13, 10115 Berlin, Germany; 3Department of Plant and Environmental Sciences, University of Copenhagen, 2630 Copenhagen, Denmark; roitsch@plen.ku.dk; 4Department of Adaptive Biotechnologies, Global Change Research Institute, Czech Academy of Sciences, 603 00 Brno, Czech Republic

**Keywords:** endophytic fungi, sugar metabolism, glycolysis

## Abstract

Endophytic fungi that colonize the plant root live in an environment with relative high concentrations of different sugars. Analyses of genome sequences indicate that such endophytes can secrete carbohydrate-related enzymes to compete for these sugars with the surrounding plant cells. We hypothesized that typical plant sugars can be used as carbon source by root endophytes and that these sugars also serve as signals to induce the expression and secretion of glycolytic enzymes. The plant-growth-promoting endophytes *Serendipita indica* and *Serendipita herbamans* were selected to first determine which sugars promote their growth and biomass formation. Secondly, particular sugars were added to liquid cultures of the fungi to induce intracellular and extracellular enzymatic activities which were measured in mycelia and culture supernatants. The results showed that both fungi cannot feed on melibiose and lactose, but instead use glucose, fructose, sucrose, mannose, arabinose, galactose and xylose as carbohydrate sources. These sugars regulated the cytoplasmic activity of glycolytic enzymes and also their secretion. The levels of induction or repression depended on the type of sugars added to the cultures and differed between the two fungi. Since no conventional signal peptide could be detected in most of the genome sequences encoding the glycolytic enzymes, a non-conventional protein secretory pathway is assumed. The results of the study suggest that root endophytic fungi translocate glycolytic activities into the root, and this process is regulated by the availability of particular plant sugars.

## 1. Introduction

*Serendipita indica* (former known as *Piriformospora indica*) and *Serendipita herbamans* are two closely related root-endophytic fungi of the order Sebacinales [1]. *Serendipita indica* was discovered in the Thar Desert, India [2,3,4,5,6], and since then, numerous studies have confirmed its ability to promote plant growth and to increase plant resistance and tolerance to biotic and abiotic stresses [5,6,7,8]. *Serendipita herbamans* was isolated from different grasses in the Black Forest, Germany, and described to have similar effects to *S. indica* [9]). In most interactions, these fungi are considered to be endophytes because they colonize plant tissues neither causing disease symptoms (as pathogens) nor forming specialized structures (as mycorrhizal fungi). It has, however, been reported that *S. indica*, similar to other Sebacinales, can also form orchid and ericoid mycorrhiza [10,11]. One central aspect of mycorrhizal symbioses, and probably also of non-mycorrhizal interactions of plant roots, is the exchange of nutrients. Such root-colonizing fungi facilitate the uptake of minerals by the plant, receiving carbohydrates in exchange (reviewed by [12]). This nutrient exchange is well investigated in arbuscular mycorrhizal (AM) symbiosis. AM fungi forming arbuscules in the apoplast of root cortex cells are surrounded by the plant-derived periarbuscular membrane. Together with the fungal arbuscular membrane, it acts as an exchange interface between the plant and the fungus. Plant and fungal membranes carry a number of nutrient transporters such as for sugars, phosphate and ammonium [13,14,15,16]. Several genes for hexose and disaccharide transporters have been also predicted in the *S. indica* genome [17], and some of them were characterized by Rani et al. [18].

In plants, sucrose is the main transported carbohydrate from source (photosynthetic active leaves)-to-sink tissues (all other organs including roots). During their association with plants, root-colonizing fungi can be considered as additional sink organs influencing the partitioning of sugars and altering the photosynthetic rate [19,20,21]. In the roots, sucrose is cleaved in the apoplast by the cell wall invertase (CWI), forming glucose and fructose. Glucose in addition to sucrose and fructose is the primary carbon source used by most organisms. These sugars can also act as signal molecules [22,23]. 

D-mannose is another important plant sugar, and it occurs as a component of cellulose, hemicellulose, or bound to other compounds forming glycoproteins (mannoproteins) and glycolipids. In some plant residues, it can reach 12% of the dry weight [24]. Mannose is a common component of plant carbohydrate polymers for both structural and storage uses and presents a ubiquitous element of the plant secondary cell wall and the main component of the neutral fraction of the polysaccharide matrix. It is almost not present as free sugar, except after the degradation of storage polymers. Mannose is also important in fungal cell wall physiology as glycosylated proteins, called mannoproteins or mannans [25]. Mannose enter the glycolysis through hexokinase activity, forming mannose-6-phosphate that is further converted to fructose-6-phosphate by the enzyme phosphomannose isomerase.

Additionally, other sugars such as arabinose, xylose and galactose, are essential components of the plant cell. They are plant-specific, constituting an important percentage of the sugar residues in the cell wall, mainly forming hemicellulose (mainly arabinoxylans). These sugars enter in glycolysis through intermediates: arabinose through fructose-6-phosphate, galactose through glucose-1-phosphate and xylose through fructose-1,6-bisphosphate [26,27]. As they are widespread in tissues colonized by fungal endophytes, they could have a role as a carbon source or in signaling during plant–fungus interaction.

Glycolysis is the pathway through which cells metabolize glucose. Hexokinase (HK), the first step of glycolysis, forms glucose-6-phosphateas, a negatively charged intermediate that due to its net charge cannot diffuse through the cell membrane. HK performs 6-phosphorylation not only of glucose, but also of fructose, mannose and D-glucosamines [28,29]. This phosphorylation induces an electrical destabilization of the molecule needed for the second step of the pathway. Glucose-6-phospate turns to fructose-6-phosphate through an isomerization step. This transition from an aldose into a ketose catalyzed by a phosphoglucose isomerase (PGI) creates the substrate for the following enzyme. Phosphofructokinase (PFK) sets the pace of the glycolytic rate and catalyzes the transfer of a second phosphate group forming fructose 1,6-bisphosphate, which marks the end of the first stage of glycolysis. In the following step, an aldolase cleaves the six-carbon molecule into two molecules of three carbons each, dihydroxyacetone phosphate (DHAP) and glyceraldehyde 3-phosphate (GAP), completing the second stage of glycolysis. 

The conventional protein secretion (CPS) pathway is very well documented in fungi and it is characterized by a N-terminal signal peptide for translocation from the endoplasmic reticulum (ER) to the Golgi apparatus and the extra-cellular secretion-mediating vesicles [30]. The signal peptide is used by the fungal cell to recognize the proteins which should be directed towards the ER membrane and is cleaved off after translocation of the protein into the ER. The presence of this signal peptide is characteristic for proteins leaving the cell via the CPS. Based on secretome studies, many proteins have been found to be secreted in the external medium without carrying a signal peptide, and this is called the unconventional secretion pathway (USP) [31]. Different studies revealed the presence of cytoplasmic enzymes in extracellular vesicles isolated from *Candida albicans* [32] and *Saccharomyces cerevisiae* [33]. These included glycolytic enzymes such as an enolase. In *Paracoccidioides brasiliensis*, different secretory pathways were suggested as an enolase was detected in a vesicular and in a non-vesicular fraction [34]. An enolase-internal sequence of several amino acids was reported to be related to the USP in *Bacillus subtilis* [34,35,36]. In addition to this enolase, the presence of several mostly glycolytic enzymes in extracellular vesicles were reported in the plant-pathogenic genus *Paracoccidioides* [37]. This comprises, among others, glucose-6-phosphate isomerase, a triosephosphate isomerase, a glyceraldehyde-3-phosphate dehydrogenase and a hexokinase. Recently, proteins of *S. indica* axenically secreted into the medium were compared with those secreted into the apoplast of the plant [38,39]. Among these proteins, several enzymes also directly connected to glycolysis were detected.

In summary, sugars usually found in the apoplast of plants could at the same time be sources for carbohydrates and signals impacting the metabolism of root-colonizing fungi. The objectives of the current work are, first of all, revealing which plant sugars can be used by the endophytes *S. indica* and *S. herbamans* as sources for growth and development, and secondly, which of these sugars can also act as signals for regulating glycolysis inside the fungal cells and for the secretion of the corresponding enzymatic activities. The experiments were carried out in axenic cultures to avoid any bias from plant metabolic activities. First, the two endophytes were provided with different plant sugars, and growth and biomass development were monitored. Secondly, enzymes from the mycelium and the supernatant of pure cultures of *S. indica* and *S. herbamans* fed with different sugars were extracted after five days of growth to reduce any possible contamination of cytosolic enzymes coming from natural hyphal turnover. The recovered proteins were concentrated and studied through the enzymatic assay platform according to the protocol proposed by Jammer et al. [40].

## 2. Materials and Methods

### 2.1. Fungal Cultivation

*Serendipita indica* (DSM11827 [2]) and *Serendipita herbamans* (DSM 27534 [9]) were maintained on Complete medium at 28 °C [41]. To analyze fungal carbon source uptake, *S. indica* and *S. herbamans* were grown in YNB minimal medium (Merck, Darmstadt, Germany) supplemented with ammonium sulphate (2 mM) according to Zuccaro et al. [17]. This medium does not contain any nitrogen or carbon sources and thus allows us to test the utilization of different single sugars. The medium was complemented with 20 mM ammonium sulfate, 0.8% agar and 2% glucose, fructose, mannose, arabinose, xylose, galactose, mannose, sucrose, lactose, melibiose or no sugar as a negative control. Fungal growth was monitored by measuring the diameter of the colonies, and fungal biomass was determined by melting the agar for few seconds in a microwave and collecting the mycelium on a Whatman^®^ filter paper (Merck, Darmstadt, Germany) placed on a Büchner funnel connected to a vacuum pump. Mycelium collected on the Whatman^®^ filter paper was dried overnight at 65 °C. Fresh and dry mycelium weights were recorded.

To analyze fungal enzymatic activities induced by sugar application, fungi were cultivated in a complemented YNB medium without agar for 5, 10 and 14 days; lactose and melibiose were not applied. After 5, 10 and 14 days, the mycelium was separated from the culture medium by filtration with Whatman^®^ Filter Paper (Merck, Darmstadt, Germany).

### 2.2. Protein Extraction and Enzymatic Activity Analysis

To analyze internal enzymatic activities, proteins were extracted from mycelium by adapting a method used for plants [40]. Table 1 shows a schematic representation of fractions and related enzymes recovered. Mycelium was frozen in liquid nitrogen and ground in polyvinylpyrrolidone (PVPP). The amount of PVPP depends on the fungal material and oxidant capacity of the tissue. Per 0.5 g of fresh weight, 1 mL of extraction buffer (1 M NaCl, 200 mM HEPES/NaOH pH 7.5, 3 mM MgCl_2_, 1 mM EDTA, 2% glycerol, 0. 1 mM PMSF, 1 mM benzamidine [42]) was added. After centrifugation at 8000 rpm for 10 min, the crude extract was separated from the pellet. The pellet treated with a high salt buffer (1 M NaCl, 200 mM HEPES/NaOH pH 7.5, 3 mM MgCl_2_, 1 mM EDTA, 2% glycerol, 0.1 mM PMSF, 1 mM benzamidine [42].

Both cytoplasmic and external protein samples were dialyzed overnight following the protocol proposed by Jammer et al. [40] and finally centrifuged using 50 mL Amicon^®^ Ultra Centrifugal Filters (Merck, Germany) to concentrate and recover the proteins. Concentrations of proteins extracted from the mycelium and obtained from the culture filtrates were estimated using Bradford protein assay (Bio-Rad, Hercules, CA, USA). Enzymatic activities were measured according to [40]). The assay is based on differential light absorbance of NAD(P)^+^ and NAD(P)H + H^+^. NAD(P)H + H^+^ is formed during the conversion of the sugar substrate by the enzymes. The activities of aldolase, phosphofructokinase (PFK), glucose-6-phosphate dehydrogenase (G6PDH), phospho-glucose isomerase (PGI), phospho-glucose mutase (PGM), glycogen pyrophosphorylase (GP; adapted by glucose-1-phosphate adenylyl transferase method), UDP-glucose pyrophosphorylase (UGPase), acidic and neutral invertase, hexokinase (HK) and fructokinase (FK) were quantified as described above for the fungal material.

### 2.3. Protein Secretion and Signal Peptide

DNA sequences from *S. indica* are publicly available (GenBank GCA_000313545.1), whereas DNA sequences from *S. herbamans* were kindly provided by Alga Zuccaro (University of Cologne, Cologne, Germany). The sequences of the selected genes were analyzed by different open source tools for secretome analysis (PrediSi, SignalP, TargetP and PSort) to predict the secretion of the enzymes in the external medium.

### 2.4. Statistical Analysis

To analyze sugar uptake and enzymatic activities, 5 and 6 replicates of each treatment were used, respectively. To analyze external invertase activity, 3 replicates were used.

Statistical analyses were carried out using Statistica13 (StatSoft, Tulsa, OK, USA) and SigmaPlot (Systat Software Inc., San Jose, CA, USA) software. When samples were homogeneously distributed, one-way ANOVA analysis was applied to test differences between the mock and the experimental treatments, separately. To determine the homogeneous groups by post hoc analysis, Tukey’s Honestly Significant Difference (HSD) test was applied.

## 3. Results

### 3.1. Sugar Utilization

To study sugar utilization, *S. indica* and *S. herbamans* were grown on solid media where the additionally applied sugar was the only available carbon source. The amount of hyphal biomass inoculated in a single point in the center of the plate is considered to be a colony. After two weeks, colony diameters and fungal biomasses were quantified, and colony densities were calculated (Figure 1).

Fungal growth responses to the different sugars did not show a direct correlation between fungal biomasses and colony diameters. Particular carbon sources, which could not be used by the fungus for developing biomass, trigged a specific morphological response characterized by fast growth of thin and sparse explorative hyphae reflected by large colony diameters. Due to this phenomenon, colony density calculated from mg of dry weight per cm of colony diameter was used as unit for estimating the usability of carbon sources. Explorative hyphae (Figure 2) were formed by *S. indica* in the presence of lactose, melibiose, arabinose and galactose and in the absence of any sugar (negative control) and by *S. herbamans* in the presence of lactose and melibiose and in the absence of any sugar. Among the disaccharides supplied to the fungi, sucrose was easily assimilated, whereas lactose and melibiose seemed not to be suitable carbon sources. Only lactose appeared to be at least partially metabolized by *S. herbamans*. Among monosaccharides, *S. indica* metabolized glucose, fructose, xylose and mannose, whereas *S. herbamans* metabolized glucose, fructose, arabinose, xylose, galactose and mannose.

### 3.2. Enzymatic Activities in the Cytosol

Results observed at the three-time frames (5, 10 and 14 days) were similar except for the highest UGPase activity at day 14. Due to this, in Table 2 and Table 3 the results at 14 days for each sugar in comparison with the control without any carbon source are shown. The activities are given in Appendix A. The different sugars added to the media affected glycolytic enzymes activities with major differences between the two fungi. Only glucose-6-phosphate dehydrogenase was not affected by any carbon source in both fungi.

In the *S. indica* mycelium, fructokinase and UDP-glucose pyrophosphorylase activities were induced by all sugars, whereas phospho-glucose isomerase was repressed. Aldolase and phosphofructokinase activities were increased if fructose or mannose were present as carbon source, hexokinase was only affected by mannose. The activity of glycogen pyrophosphorylase was repressed by glucose and sucrose, whereas phosphoglucose mutase activity was repressed by mannose. Invertase activity showed induced activities in the presence of mannose, but repressed activities in the presence of a mixture of glucose and fructose.

Fructokinase and UDP-glucose pyrophosphorylase activities were induced by all sugars in the mycelium of *S. herbamans*. In contrast, all tested sugars repressed the activities of invertase, phosphofructokinase and of phospho-glucose isomerase.

### 3.3. Enzymatic Activities in the Culture Medium

The activity of enzymes related to glycolysis was also analyzed in the culture medium of *S. indica* and *S. herbamans* 5, 10 and 14 days after starting the cultures. Different enzymatic activities were detected at the different time points. Because hyphal turnover was observed to start at 7 days after the start of the cultures, only results obtained from the 5 days’ cultures were taken for further consideration. Table 4 and Table 5 show the induction or repression by the different sugars in comparison with the control without sugar additions; the values for the activities are shown in Appendix A.

In general, the activities of all enzymes were detected except for that of UDP-glucose pyrophosphorylase. In the culture medium of *S. indica*, fructokinase activity was repressed by glucose, fructose, by a mixture of both sugars and by sucrose and mannose. Sucrose and mannose also repressed hexokinase activity. Glucose and xylose induced the activities of glycogen pyrophosphorylase and phosphoglucose mutase, glycogen pyrophosphorylase activity was additionally induced by sucrose and by the mixture of glucose and fructose. Extracellular activities of *S. herbamans* were not as regulated by sugars. Only glycogen pyrophosphorylase and phosphoglucose mutase activities were induced by glucose, whereas invertase and phospho-glucose isomerase showed higher activities in the presence of mannose.

Invertase activity was analyzed at pH 4.5 and pH 6.8 to verify the enzyme activity in an acidic environment. Invertase activities were found to be higher at pH 4.5 than at pH 6.8, and this was independent of the sugars used as carbon source (Figure 3). Invertase showed the highest activities among all analyzed enzymes (Appendix A).

## 4. Discussion

Heterotrophic microorganisms, which are associated with plants, can cover their demand for carbon at least partially by the interaction with their hosts. For this purpose, they often need to adapt physiological processes involved in the utilization of the resources provided by the plant. In this work, growth and biomass development of the root endophytic fungi *Serendipita indica* and *Serendipita herbamans* were analyzed in the presence of typical plant sugars, and hyphae-internal and -external activities of glycolytic enzymes in response to these sugars were quantified. A number of different sugars are present in the intercellular spaces and in the apoplasts of root cells where they are principally available for endophytic microorganisms. In addition, intracellular sugars are released upon programmed cell death, which can be induced by *S. indica* [43].

### 4.1. Utilization of Plant Sugars

Several plant sugars were selected which are supposed to occur in significant amounts in roots [44]. Sucrose is transported from source tissues to roots, where it can be cleaved forming glucose and fructose. Mannose is an important sugar component of mannoproteins, celluloses and hemicelluloses. Arabinose and galactose can be derived from arabinogalactans, being highly ubiquitous in plants, and xylose from the xyloglucan backbone of the primary wall and the middle lamella [24,45]. These sugars were used as the sole carbon source in a solid medium culture of the two endophytes. The morphological analysis of the cultures revealed an intensive radial growth in the presence of some sugars and in the absence of any sugar, but low biomasses (Figure 1A). This phenotype is based on the development of so-called “runner hyphae” or “explorative hyphae” and was described for thick-walled hyphae tracking the root in the soil able to infect or colonize it [46,47]. The phenotype observed is characterized by rapid growth, a sparse mycelium and a high secretory capacity of specific enzymes. It was interpreted as a nutrient stress response where the fungi explore the surrounding environment searching for better conditions. Despite the poor biomass, such runner hyphae show a very uncommon high physiological activity and active enzymatic secretion. Similar results were obtained during glucose starvation of *S. cerevisiae* [48,49]. Taking this into account, the density of the hyphae as the ratio of biomass to colony radius was taken as a measure for the utilization of the different sugars (Figure 1C).

The results of the present study showed that monosaccharides in particular can be easily used by the two *Serendipitaceae*. Glucose, fructose, sucrose, mannose, xylose and arabinose are abundant in the plant apoplast [50,51,52,53] and could thus serve as carbon sources when the two fungi are colonizing the root. Despite its presence in plant tissues [54], galactose is well metabolized only by *S. herbamans*. Hexose transporters have been isolated and characterized from fungi of the order Glomeromycotina, which form mutualistic symbioses with plants, and they were able to translocate many sugars including galactose across membranes [55,56]. Sequences for hexose transporters can be also detected in the genomes of the two *Serendipitaceae* and one of the genes from *S. indica* highly induced during colonization of the root has been characterized [18]. Glucose transport could be outcompeted by a number of sugars, but barely by galactose, supporting the results of the present study. However, a final assessment needs the analyses of all putative genes expressed in the intraradical mycelia of both *Serendipitaceae*. The ability to use galactose as carbon source is usually common among yeasts and fungi [57], but this capability is impaired in *S. indica*. Interestingly, the grass endophyte *Acremonium* sp., was also shown to be unable to metabolize this sugar [58]. This inability might be based on a lack of this sugar in the natural hosts of both *Acremonium* sp. and *S. indica*, but it needs further investigations with endophytes and their hosts to confirm this assumption.

Among the disaccharides, only sucrose resulted in the significant formation of biomasses. Sucrose as the main transport sugar is present in high concentrations in the apoplast and can be used as a carbon source by many plant-interacting fungi [59,60,61]. This could be based on the uptake of the disaccharide and intracellular metabolization or by the secretion of an invertase and the subsequent uptake of glucose and fructose (see discussion below). Lactose can be very well metabolized by saprotrophic and pathogenic fungi [62], but its utilization is neglectable in both *Serendipitaceae*. This could be interpreted as a lack of enzymes able to break the chemical bounds in lactose. Melibiose such as lactose needs an alpha-galactosidase to cleave the molecule in glucose and galactose. The two disaccharides only differ in the chirality of the ring formed by the galactose [63]. This similarity could explain why both cannot be metabolized because the endophytes do not harbor alpha-galactosidase activities, despite the fact that corresponding sequences are present in the genome of *S. indica.* Lactose and melibiose are only present in low concentrations in roots; therefore, their utilization as carbon sources might not be needed in addition to the other highly abundant sugars.

### 4.2. Cytoplasmic Activity of Glycolytic Enzymes

Sugar availability modifies the pacing rate of glycolysis, and the function of particular sugars as signals for regulating the expression of corresponding genes and the activity of glycolytic enzymes has been investigated in a number of studies (see below). Sugars, which are not directly catabolized in glycolysis, such as arabinose, galactose and xylose, enter the glycolytic pathway through glucose and fructose intermediates (fructose-6-phospate, glucose-1-phosphate and fructose-1,6-bisphosphate, respectively). For this reason, their effect as signals has not been analyzed in the current study. In the following discussion, starvation conditions (no sugar) have been taken as baseline. Although this is an extreme situation of metabolic stress, it shows how particular sugars act as signals on the carbohydrate metabolism (Table 1 and Table 2).

The analysis of the glycolytic pathway shows that common plant-derived sugars can regulate the activity of different enzymes without being their direct substrates. Concerning their regulation in *S. indica* (Table 2), sugars can be divided into two groups. Fructose and mannose have mainly activating effects, whereas glucose, sucrose and the mixture of both inhibit or activate particular enzymatic activities. In contrast, the different sugars showed nearly identical effects in *S. herbamans* (Table 3). The variation in the patterns can currently not be explained, but might be based on the different natural ecosystems and hosts where the endophytes have been isolated, and/or by differences in the mode of colonization and induction of programmed cell death. Both fungi, however, showed an induction of fructokinase and UDP-glucose pyrophosphorylase and a very strong inhibition of phosphoglucose isomerase activity compared with starving conditions. Fructose or glucose-1-phospate (coming from glycogen or starch) are not available at starving conditions; therefore, enzymes such as fructokinase and UDP-glucose pyrophosphorylase are not required. In contrast, phosphoglucose isomerase activities are extremely high under starving conditions which might be due to the need that any trace of glucose-6-phosphate (G6P) must be converted to fructose-6-phosphate (F6P) to feed the further steps of glycolysis.

### 4.3. Extracellular Activities of Glycolytic Enzymes

Studies of *S. indica* reported the secretion of numerous proteins in the interaction with the plant [38,39]. Among many other proteins, Thürich [39] identified an invertase and a phosphoglucose isomerase in the plant-induced secretome, whereas [38] also found these two enzymes plus a glycogen pyrophosphorylase, a phosphoglucose mutase and a fructokinase. Very recently, ref. [64] detected the secretion of several glycosyl hydrolases, glycosyl phosphorylases and invertase in *S. indica*. In contrast to these studies, we did not identify particular proteins with induced secretion in the presence of the plant, but measured extracellular enzymatic activities and analyzed if particular plant sugars induce these activities.

Among the enzymes, where activities have been detected in the culture supernatant, the invertase holds an exceptional position. Both fungal genomes harbor only one gene. Based on the sequence, the presence of a signal peptide for secretion could be predicted for *S. indica*, but not for *S. herbamans*. This suggests the presence of an unconventional secretory pathway which will be discussed below. Under starvation conditions, the extracellular activities were already high in *S. indica* (Table 4) and extremely high in *S. herbamans* (Table 5), and these activities were even further induced by particular sugars. An interesting characteristic of the invertase of both endophytes was the fact that its activity was seven times higher at pH 4.5 than at pH 6.8 (Figure 3A,B). In plant-interacting fungi, this feature could be an adaptation to the endophytic lifestyle, as the pH of plant apoplasts is in the range of around 5 [65,66]. Additionally, in the pathogenic fungus *Puccinia graminis*, the optimal activity of invertase is at pH 5.5 [67]. Extracellular invertase activity was not only induced by sucrose as expected, but also by an equimolar mix of glucose and fructose, showing the opposite effect as described in plants and in *S. cerevisiae*, where high concentrations of glucose and fructose have an inhibiting effect [68,69,70]. This could be explained by the different metabolic functions of the enzyme in fungi and in yeast or plants. Although invertase is needed in yeast and plants to balance the stock of sugars, in fungi it is used to mobilize as many sugars as possible.

Besides invertase, both fungi showed significant extracellular activities for all tested enzymes except for the UDP-glucose pyrophosphorylase. With these enzymes, both fungi can catalyze the turn-over of sugars to glyceraldehyd-3-phosphate (Figure 4). Both fungi also show an induction of glycogen pyrophosphorylase and phosphoglucose mutase by glucose (Table 4 and Table 5). Glycogen pyrophosphorylase polymerizes glycogen from glucose-1-phosphate which could be provided by the activity of the phosphoglucose mutase catalyzing the conversion of glucose-6-phosphate to glucose-1-phosphate. Therefore, it is possible that both fungi use the availability of glucose for the synthesis of glycogen outside of their cells. Secretion of glycogen pyrophosphorylase has been also reported for *Paracoccidioides brasiliensis*, a human pathogen, without knowing its role in pathogenesis [71].

Fructokinase activities are reduced in *S. indica* when “easy to metabolize” sugars are present, such as glucose, fructose and sucrose. Mannose also reduces fructokinase activity, and in this respect it is important to know that the phosphorylation of mannose by fructokinase has been shown as a side activity in different organisms such as *Escherichia coli* and spinach [72,73]. External activity of fructokinase was described for the wood-degrading species *Chaetomium globosum*, where it is also inhibited by mannose [74,75]. The secretion of phosphoglucose isomerase was also described in *Paracoccidioides* sp. [37,71], as mentioned above for the glycogen pyrophosphorylase.

In summary, the inhibition of the external activities of particular enzymes in *S. indica* can be inversely interpreted as a starvation response when no sugars are present. Why they are not reduced in *S. herbamans* upon the application of sugars can, however, not be explained without knowing more about the biology of the endophytes in their different natural environments.

### 4.4. Secretion

The detection of extracellular activities in the present study and of proteins in the studies by Thürich et al. [39] and Nizam et al. [38] suggests a high secretion activity in the two endophytic fungi. Except for the invertase of *S. indica*, no enzyme seems to have a conventional signal peptide for secretion according to the genome sequences, and apoplastic occurrence could not be predicted in silico. The absence of conventional signal peptides for secretion is a well-known phenomenon. Most proteins, which were secreted in *A. thaliana* after stimulation with silicic acid or pathogens, did not have an N-terminal signal peptide [76,77,78]. A C-terminal microbody targeting signal (CMTS) residing in the last three amino acids of the sequence (-NRA) has been detected in the sequence of the phosphoglucose isomerase. This secretion system was proven in many organisms (plant, yeast, insects and mammals) and involves the transport of the enzyme through microbodies. Microbodies are small organelles with a single membrane, including peroxisomes, glyoxysomes, and glycosomes able to translocate and secrete proteins [79,80,81].

Comparing cytoplasmic and extracellular activities, a correlation has been found for the phosphoglucose mutase where the activity similarly increases outside and inside of the hyphae in the presence of glucose. In contrast, inverse correlations were detected for invertase, fructokinase and hexokinase activities in *S. indica* and for phosphofructokinase in *S. herbamans*. This implies that secretion is regulated by sugar signals and that this is not general, but specific for particular enzymes. To the best of our knowledge, this is the first case where such an enzyme-specific secretion by a sugar has been detected.

### 4.5. A New Holobiontic Model

Based on the present results and on the glycolytic enzymes detected by Thürich et al. [39] and Nizam et al. [38] in the apoplast of *S. indica*-colonized plants, we postulate a new concept concerning processes at the plant–fungus interface during their interaction. According to this concept, this interface is not only used for communication and exchange of nutrients, but also allows active primary metabolism reactions in a shared environment between the two organisms. The enzymatic reactions shown in the present study allow the conversion of hexoses and sucrose into dihydroxyacetonphosphate and glyceraldehyd-3-phosphate (Figure 4). Considering the enzymes detected by Nizam et al. ([38] and personal communication), a non-interrupted pathway is present in the extracellular plant–fungus interface until the formation of pyruvate. Pyruvate could be easily taken up, because pyruvate transporters, which are typically present in the mitochondrial membrane, can be also incorporated into the cell membrane [82].

In addition to nutrition, external glycolysis could also be part of a strategy to decrease plant defense reactions. *S. indica* and *S. herbamans* induce sucrose biosynthesis in tomato roots (publication in preparation). Because high sucrose/hexose ratios trigger defense mechanisms such as the production of phytoalexins [83], the rapid removal of the newly synthesized sucrose in the apoplast could be important to decrease the defense response of the plant. In addition, the glycolytic pathway would lead to an accumulation of extracellular ATP (eATP) in the apoplast of roots colonized by *S. indica* and this accumulation has been confirmed [38]. Such an increase in eATP can lead to a suppression of the expression of defense-related genes, as ATP can work as negative regulator of defense signal transduction [84].

This interface would be an integrated holobiontic cytosol able to sustain an energy-producing system with the regeneration of ATP and NAD+/H through the glycolytic pathway. Once these cofactors are actively regenerated, they could also sustain protein synthesis systems. This would imply the presence of ribosomal proteins in the apoplast. So far, reports of ribosomes in the apoplast are not known, although Kwon et al. [85] reported the presence of 60S ribosomal protein L12 and the elongation factor 1-α in *A. thaliana* apoplast extract, but it was considered by the authors as cytosolic leakage. Nizam et al. ([38] and personal communication) reported the presence of fungal protein PIIN_04876 in co-cultivation with *S. indica* in a barley apoplast, with probable TIF2 translation initiation factor eIF4A, PIIN_06177 related to translation elongation factor eEF1, and bioinformatics analysis predicted the secretion of PIIN_03008, probable translation elongation factor eEF _1. Long distance transports through the phloem of several species of RNA is known in plants [86], and this could support the presence of ribosomal complexes or tRNA for an eventual apoplastic protein synthesis.

## 5. Conclusions

In summary, typical plant sugars can modulate the secretion of fungal cytosolic enzymes. The first part of the glycolytic pathway was recorded and analyzed in this work, and other studies suggest that the entire pathway may be present. The activity of the secreted enzymes is unchanged. These findings lead to the hypothesis of external glycolysis in the apoplast converting it into an additional compartment where sugars can be metabolized in a closed space before the resulting metabolites are transported to the cytoplasm.

## Figures and Tables

**Figure 1 microorganisms-10-00320-f001:**
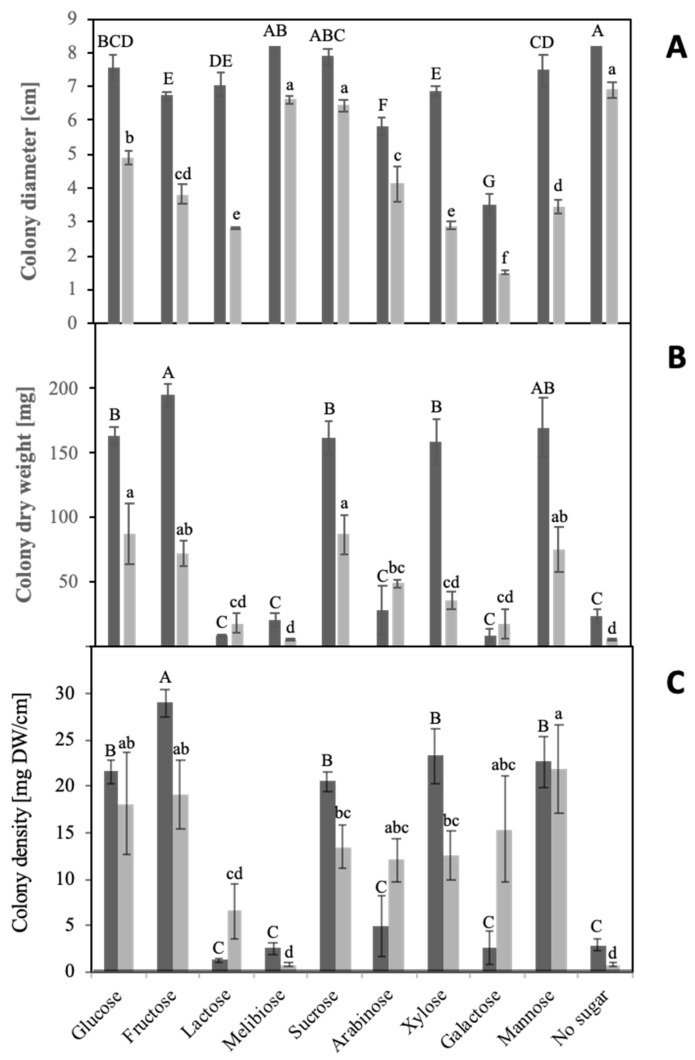
Growth patterns. *Serendipita indica* (dark grey columns) and *Serendipita herbamans* (light grey columns) were grown on Petri plates for 14 days with different sugar sources. Colony diameters (**A**) and dry weights (**B**) were measured, and colony densities (**C**) were calculated. Mean values are shown with standard errors as bars. Significance of differences of growth parameters between different sugar sources were analyzed for the two fungi independently and are indicated by different lower- or upper-case letters, respectively (HSD Tukey Test; *n* = 5).

**Figure 2 microorganisms-10-00320-f002:**
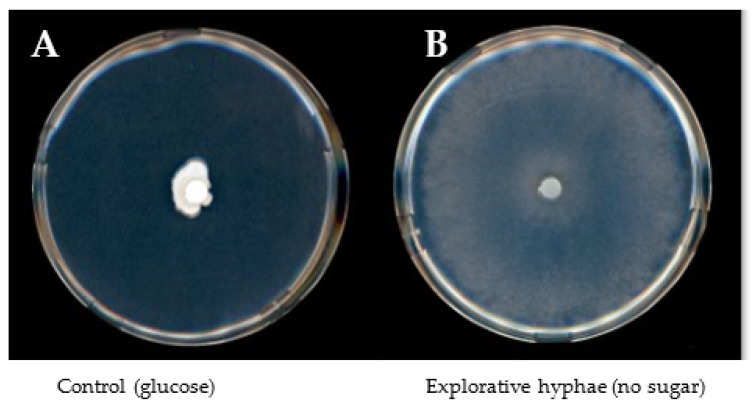
*S. indica* growing on glucose (**A**) and explorative hyphae (**B**) 7 days after inoculation. Explorative hyphae are characterized by fast growth, thin and sparse mycelium appearing when *S. indica* and *S. herbamans* are cultivated in a medium lacking a suitable carbon source.

**Figure 3 microorganisms-10-00320-f003:**
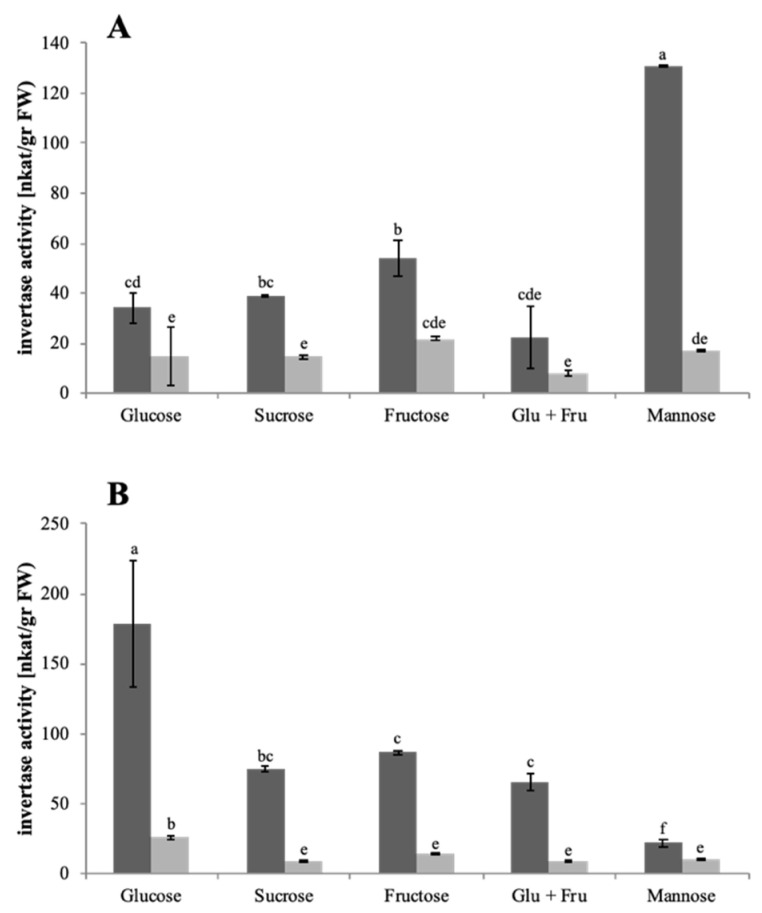
Extracellular invertase activities. *S. indica* (**A**) and *S. herbamans* (**B**) were grown in liquid cultures with different sugar sources. Culture supernatant was harvested, proteins were extracted and invertase activities were measured at pH 4.5 (dark grey columns) or at pH 6.8 (light grey columns). Mean values are shown with standard errors as bars. Significant differences of growth parameters between different sugar sources and different pH are indicated by different letters above the columns (two-way ANOVA Tukey Test; *n* = 3).

**Figure 4 microorganisms-10-00320-f004:**
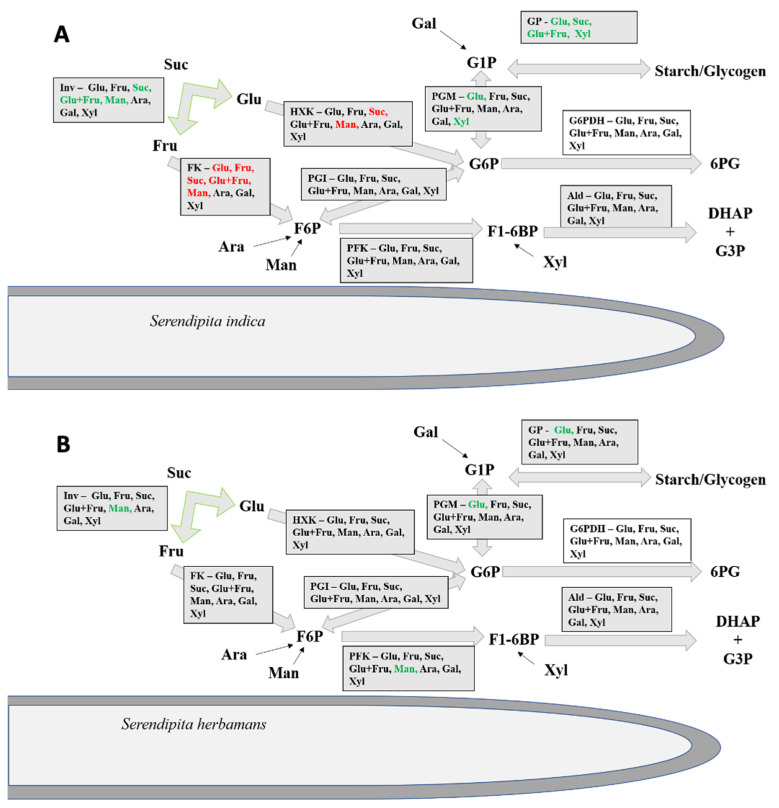
Extracellular carbohydrate metabolism. Model illustrating the fungal secretion of enzymes related with glycolysis. Figure represents the metabolic reactions found secreted in the external medium by *S. indica* (**A**) and *S. herbamans* (**B**). Plant sugars (glucose: Glu, fructose: Fru, sucrose: Suc, an equimolar mix of glucose and fructose: Glu+Fru, mannose: Man, arabinose: Ara, galactose: Gal, xylose: Xyl) induce (green) or repress (red) the activities of specific enzymes compared with the activities of cultures grown without carbon source. Inv: invertase; FK: fructokinase; HXK: hexokinase; G6PDH: glucose-6-phosphate dehydrogenase; GP glycogen pyrophosphorylase; PFK: phosphofructokinase; PGI: phosphoglucose isomerase; PGM: phosphoglucose mutase; UGPase: UDP-glucose pyrophosphorylase; Ald: aldolase. Suc: sucrose: Glu: glucose; Fru: fructose; Ara: arabinose; Man: mannose; Gal: galactose; Xyl: xylose; G1P: glucose-1-phosphate; G6P: glucose-6-phosphate; F6P: fructose-6-phosphate; F1-6BP: fructose-1-6-bisphosphate; 6PG: 6-Phosphogluconolacton; DHAP: dihydroxyacetonphosphate; G3P: glyceraldehyd-3-phosphate.

**Table 1 microorganisms-10-00320-t001:** Schematic representation of fractions after disruption of cells and centrifugation used for the analysis of enzyme activities.

Sample Fraction		Enzyme
Supernatant	Centrifuged raw extract	aldolase, phosphofructokinase, glucose-6-phosphate dehydrogenase, phosphoglucose isomerase, phospho-glucose mutase, UDP-glucose pyrophosphorylase
	Dialyzed	vacuolar (acidic) invertase, cytosolic (neutral) invertase, hexokinase and fructokinase

**Table 2 microorganisms-10-00320-t002:** Sugar regulation of enzyme activities in the cytosol of *S. indica* 14 days cultivated in the presence of glucose (Glu), fructose (Fru), sucrose (Suc), an equimolar mix of glucose and fructose (Glu + Fru), mannose (Man) or without sugar. The table shows the M-values (log2 of ratios of values obtained from sugar-treated cultures to values from cultures cultivated without sugar). Values higher than 1 (>2-fold induced) below −1 (>2-fold repressed) and which are based on significant differences (Tukey HSD test; *n* = 6) are indicated in bold and color-coded (green: sugar induction; red: sugar repression). Activity of glucose-6-phosphate dehydrogenase was very low and not affected.

*S. indica*	Glu	Fru	Suc	Glu + Fru	Man
Aldolase	−0.35	**1.42**	−0.03	−0.35	**1.3**
Fructokinase	**1.86**	**3.08**	**2.36**	**1.82**	**3.58**
GP	**−1.32**	0.79	**−3.91**	−0.91	0.85
Hexokinase	−0.16	0.68	−0.03	−0.35	**1.55**
Invertase	−0.56	0.1	−0.37	**−1.17**	**1.38**
PFK	−1.58	**2.12**	0.85	0.42	**4.39**
PGI	**−5.45**	**−6.03**	**−4.91**	**−5.23**	**−6.26**
PGM	0.07	−0.46	0.04	−0.31	**−1.07**
UGPase	**1.14**	**2.7**	**1.85**	**1.85**	**2.81**

GP: glycogen pyrophosphorylase; PFK: phosphofructokinase; PGI: phosphoglucose isomerase; PGM: phosphoglucose mutase; UGPase: UDP-glucose pyrophosphorylase.

**Table 3 microorganisms-10-00320-t003:** Sugar regulation of enzyme activities in the cytosol of *S. herbamans* 14 days cultivated in the presence of glucose (Glu), fructose (Fru), sucrose (Suc), an equimolar mix of glucose and fructose (Glu + Fru), mannose (Man) or without sugar. The table shows the M-values (log2 of ratios of values obtained from sugar-treated cultures to values from cultures cultivated without sugar). Values higher than 1 (>2-fold induced) below −1 (>2-fold repressed) and which are based on significant differences (Tukey HSD test; *n* = 6) are indicated in bold and color-coded (green: sugar induction; red: sugar repression). Activity of glucose-6-phosphate dehydrogenase was very low and not affected.

*S. herbamans*	Glu	Fru	Suc	Glu + Fru	Man
Aldolase	−2.26	−3.35	−2.65	−3.34	−2.18
Fructokinase	**3.95**	**3.23**	**3.58**	**3.46**	**3.88**
GP	−0.34	−2.24	1.98	−2.26	−0.54
Hexokinase	0.38	−0.15	0.32	−0.14	0.13
Invertase	**−1.79**	**−3.05**	**−2.84**	**−3.24**	**−4.82**
PFK	**−2.55**	**−2.47**	**−2.31**	**−3.44**	**−3.34**
PGI	**−7.35**	**−6.54**	**−6.59**	**−7.01**	**−9.90**
PGM	0.02	0.72	0.53	0.26	−0.95
UGPase	**2.90**	**1.80**	**5.05**	**3.73**	**5.07**

GP: glycogen pyrophosphorylase; PFK: phosphofructokinase; PGI: phosphoglucose isomerase; PGM: phosphoglucose mutase; UGPase: UDP-glucose pyrophosphorylase.

**Table 4 microorganisms-10-00320-t004:** Sugar regulation of extracellular enzyme activities of *S. indica* 5 days cultivated in the presence of glucose (Glu), fructose (Fru), sucrose (Suc), an equimolar mix of glucose and fructose (Glu + Fru), mannose (Man) or without sugar. The table shows the M-values (log2 of ratios of values obtained from sugar-treated cultures to values from cultures cultivated without sugar). Values higher than 1 (>2-fold induced) below −1 (>2-fold repressed) and which are based on significant differences (Tukey HSD test; *n* = 6) are indicated in bold and color-coded (green: sugar induction; red: sugar repression). Activities of UDP-glucose pyrophosphorylase were not detected.

*S. indica*	Glu	Fru	Suc	Glu + Fru	Man	Ara	Gal	Xyl
Aldolase	−0.60	−1.39	−1.72	−2.16	0.26	−0.18	−0.55	−0.49
Fructokinase	**−1.38**	**−2.03**	**−1.91**	**−1.71**	**−2.48**	−0.59	−0.75	−0.62
G6PDH	−1.10	−1.30	−1.23	−0.99	−1.00	−0.99	−0.04	−1.01
GP	**16.28**	n.d.	**11.81**	**13.81**	n.d.	n.d.	n.d.	**16.33**
Hexokinase	−1.45	−1.21	**−1.80**	−1.66	**−1.95**	−0.29	−0.07	−0.32
Invertase	1.79	1.75	**3.33**	**3.73**	**3.34**	−1.07	−0.17	−0.20
PFK	−0.95	−1.87	−2.00	−2.10	−0.52	−0.52	−0.35	−0.63
PGI	−3.07	−1.57	−1.59	−2.32	−0.53	0.63	0.03	−1.96
PGM	**2.49**	−0.75	−0.69	0.54	0.95	0.07	−1.49	**2.43**

G6PDH: glucose-6-phosphate dehydrogenase; GP: glycogen pyrophosphorylase; PFK: phosphofructokinase; PGI: phosphoglucose isomerase; PGM: phosphoglucose mutase.

**Table 5 microorganisms-10-00320-t005:** Sugar regulation of extracellular enzyme activities of *S. herbamans* 5 days cultivated in the presence of glucose (Glu), fructose (Fru), sucrose (Suc), an equimolar mix of glucose and fructose (Glu + Fru), mannose (Man) or without sugar. The table shows the M-values (log2 of ratios of values obtained from sugar-treated cultures to values from cultures cultivated without sugar). Values higher than 1 (>2-fold induced) below −1 (>2-fold repressed) and which are based on significant differences (Tukey HSD test; *n* = 6) are indicated in bold and color-coded (green: sugar induction; red: sugar repression). Activities of UDP-glucose pyrophosphorylase were not detected.

*S. herbamans*	Glu	Fru	Suc	Glu + Fru	Man	Ara	Gal	Xyl
Aldolase	1.93	2.16	1.90	1.86	2.45	1.30	1.18	1.67
Fructokinase	1.15	0.90	0.70	0.91	1.04	0.92	0.56	0.71
G6PDH	0.26	0.81	−0.62	−0.74	0.63	−0.86	−1.74	−1.32
GP	**6.17**	2.00	4.31	4.57	3.67	0.49	1.32	4.83
Hexokinase	0.85	1.07	0.34	0.39	1.34	0.68	0.32	0.92
Invertase	−0.41	1.15	1.01	1.05	**1.33**	−0.28	−0.06	0.65
PFK	1.42	1.22	1.18	1.09	**1.87**	0.79	0.41	0.96
PGI	−0.69	0.88	−0.28	−0.37	0.69	−0.24	−0.37	−0.67
PGM	**4.28**	2.64	3.02	2.75	2.55	1.58	0.81	3.34

G6PDH: glucose-6-phosphate dehydrogenase; GP glycogen pyrophosphorylase; PFK: phosphofructokinase; PGI: phosphoglucose isomerase; PGM: phosphoglucose mutase.

## Data Availability

Not applicable.

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
