# Peer review of "Extracellular Glycolytic Activities in Root Endophytic Serendipitaceae and Their Regulation by Plant Sugars"

_microorganisms, 2022, doi:10.3390/microorganisms10020320_

Round 1
Reviewer 1 Report
The article is about testing the growth properties of two endophytic fungi on different sugars. This is followed by measurement of enzymatic activities of sugars reducing enzymes in fungi biomass and in media to make some comparison between the ability to metabolize particular sugar and enzymatic activity. Based on this authors made some conclusions about the possible role of sugars in “communication” between plant and its root colonizing endophytic fungi. The promised aim was meet.
What resulted in “minor revision needed” is next to small mistakes problem of cultivation time. The growth of fungi on different sugars was on a solid medium for 14 days. This is only one what is clear. Growth of fungi for determination of mycelial enzymatic activities in liquid media was for 5, 10, and 14 days (line 143) or just for 5 and 14 days (lines 144, 150, and 222) or only for 5 days (line 224)? This is also connected with data presented in tables 2 and 3 are they after 5 days (written in lines 224 and 225) or after 14 days written in text before both tables (lines 240 and 248)? The same is with activities measured in the medium. Was it measured after 5, 10, and 14 days (line 258) or just after 14 days (text before tables 4 and 5 lines 265 and 273)? This should be clarified.
Second problem is in material and methods section. Text in lines 147-152 is in short what is again but differently written in lines 154-157. This should be rewritten to have the whole procedure written once and not in two versions which must be combined to find out how the extraction was done.
Small problems are:
Line 106 UPS should be USP
Line 155 plant tissue instead of fungal tissue
Line 175 plant material why is not there fungal?
Line 198 names of fungi should be in italic
Line 240 name of fungus should be in italic
Lines 344-346 sentence should be rewritten
Lines 437-444 belongs to picture not part of text
Author Response
“What resulted in “minor revision needed” is next to small mistakes problem of cultivation time. The growth of fungi on different sugars was on a solid medium for 14 days. This is only one what is clear. Growth of fungi for determination of mycelial enzymatic activities in liquid media was for 5, 10, and 14 days (line 143) or just for 5 and 14 days (lines 144, 150, and 222) or only for 5 days (line 224)? This is also connected with data presented in tables 2 and 3 are they after 5 days (written in lines 224 and 225) or after 14 days written in text before both tables (lines 240 and 248)? The same is with activities measured in the medium. Was it measured after 5, 10, and 14 days (line 258) or just after 14 days (text before tables 4 and 5 lines 265 and 273)? This should be clarified.”
reply:Thank you very much to point it out. I hope I made it more clear.
“Second problem is in material and methods section. Text in lines 147-152 is in short what is again but differently written in lines 154-157. This should be rewritten to have the whole procedure written once and not in two versions which must be combined to find out how the extraction was done.”
reply:Fixed.
“Small problems are: Line 106 UPS should be USP Line 155 plant tissue instead of fungal tissue Line 175 plant material why is not there fungal? Line 198 names of fungi should be in italic Line 240 name of fungus should be in italic Lines 344-346 sentence should be rewritten Lines 437-444 belongs to picture not part of text”
reply: Fixed.
Reviewer 2 Report
Dear Authors,
congratulations for your work. Please check some minor comments:
Line 126, 164 etc – name the authors when using formulas like “the protocol of..”/”according to..”
Line 222- rephrase the redundant expression; ex. Mycelia..from liquid cultures were harvested after 5 and 14 days.
Line 223 – “Tab.” Should be “Table x” in text (check carefully the instructions)
Conclusions – this section is intended to be a take away message, besides underlying major findings and future prospects. In the present state, it is more a debate/dissertation area so, it might confuse the reader as by far each step was carefully explained, the hypotheses were logically planned and the red thread conveyed the reader throughout the text. In my opinion, the most part of this “chapter” should be moved where it corresponds – discussion.
Finally, there are no such needs, nor habit to use references and indications to figures in conclusions…
Bear in mind that if you stick to your most important proposals like the “new concept” on processes at the plant fungus interaction, by only shortly describing the features you infer (i.e. primary metabolism reactions -> nutrition + plant defense), you might get more attention than you think. The means should be described in earlier steps (i.e. discussion).
Author Response
“Line 126, 164 etc – name the authors when using formulas like “the protocol of..”/”according to..” 3 Line 222- rephrase the redundant expression; ex. Mycelia..from liquid cultures were harvested after 5 and 14 days. Line 223 – “Tab.” Should be “Table x” in text (check carefully the instructions)”
reply: Fixed.
“Conclusions – this section is intended to be a take away message, besides underlying major findings and future prospects. In the present state, it is more a debate/dissertation area so, it might confuse the reader as by far each step was carefully explained, the hypotheses were logically planned and the red thread conveyed the reader throughout the text. In my opinion, the most part of this “chapter” should be moved where it corresponds – discussion.”
reply:Thank you very much for the suggestions, we did as you proposed.
“Finally, there are no such needs, nor habit to use references and indications to figures in conclusions…”
reply:Fixed.
Reviewer 3 Report
The manuscript by Rocchis et al has attempted to analyze the role of different carbon sources (sugars) from plants on the growth and biomass development of the root endophytic fungus Serendipita indica and Serendipita herbamans. Authors have also analyzed the activities of glycolytic enzymes in response to different sugars. It is a nice preliminary work on the well-known and well-favored source of carbon utilization in endophytic fungus. This study will open a new vista to explore more on sugar metabolism in root endophytic fungi. Although the manuscript is well written and discussed, a few minor improvements would be required before acceptance.
Authors can add a few lines of major findings in the abstract section like which carbon source was more favored and its associated regulation of enzyme activities in the cytosol as well as regulation of extracellular enzyme activity.
I also noticed authors have used several old references in the introduction and discussion section, however, lots of new studies have been done on AMF as well as on root endophytic fungi including P. indica and S. herbamans. I would recommend to update with recent citations.
Since authors have described about root-colonizing fungi facilitate the uptake and nutrients exchange minerals by the plant and fungus can also add recent citations and studies on P. indica signifying recent advancement related to transport and metabolism to make a wide view on the fungus. For instance, authors can add citations related to nutrients like Sulfur (Narayan et al., 2021, Plant cell), Iron (Verma et al., 2021, Environmental Microbiology), Magnesium (prasad et al., 2019 Frontiers in Microbiology) their uptake and transfer to the host plants by P. indica.
Information on some specific properties of both fungi can be added. For instance, information on the benefits of root endophytic fungus-like P. indica and S. herbamans over AMF/mycorrhizal fungi can be added.
Authors could explain what is the meaning of “colony” “colony diameter” and “colony densities”? I would rather suggest using “fungal hyphae”.
Did they also measure the biomass under broth medium?
Authors can replace figure # 2 with good quality without glare. And, in the control figure, which sugar was used? mix or any specific?
Figure #4 looks confusing. It would be better to improve figure #4 to make it more self-explanatory.
Any specific reason to measure activity at two specific pH 4.5 and 6.8 than at several pHs in a range?
Some typos are still present. Example, line 157, 159.
The name of the organism is not italicized at some places like 198, 240 as well as in the reference section.
Table #1 looks messy. So please make it more evident.
Author Response
References update and proposed references by the reviewer
reply:Thank you very much to point it out. I have update all relevant reference (see last version).
Authors could explain what is the meaning of “colony” “colony diameter” and “colony densities”? I would rather suggest using “fungal hyphae”.
reply:Fixed
Did they also measure the biomass under broth medium?
reply:Fixed
Authors can replace figure # 2 with good quality without glare. And, in the control figure, which sugar was used? mix or any specific?
reply:Fixed
Any specific reason to measure activity at two specific pH 4.5 and 6.8 than at several pHs in a range?
reply:Fixed
Some typos are still present. Example, line 157, 159. The name of the organism is not italicized at some places like 198, 240 as well as in the reference section.
reply:Fixed